# Limitation by a shared mutualist promotes coexistence of multiple competing partners

Sarah P. Hammarlund[1,2], Tomáš Gedeon [3], Ross P. Carlson [4] & William R. Harcombe [1,2 ✉]

Although mutualisms are often studied as simple pairwise interactions, they typically involve complex networks of interacting species. How multiple mutualistic partners that provide the same service and compete for resources are maintained in mutualistic networks is an open question. We use a model bacterial community in which multiple 'partner strains' of *Escherichia coli* compete for a carbon source and exchange resources with a 'shared mutualist' strain of *Salmonella enterica*. In laboratory experiments, competing *E. coli* strains readily coexist in the presence of *S. enterica*, despite differences in their competitive abilities. We use ecological modeling to demonstrate that a shared mutualist can create temporary resource niche partitioning by limiting growth rates, even if yield is set by a resource external to a mutualism. This mechanism can extend to maintain multiple competing partner species. Our results improve our understanding of complex mutualistic communities and aid efforts to design stable microbial communities.

[1] Department of Ecology, Evolution, and Behavior, University of Minnesota, St. Paul, MN, USA. [2] BioTechnology Institute, University of Minnesota, St. Paul, MN, USA. [3] Department of Mathematical Sciences, Montana State University, Bozeman, MT, USA. [4] Department of Chemical and Biological Engineering, Center for Biofilm Engineering, Montana State University, Bozeman, MT, USA. ✉email: harcombe@umn.edu

Mutualisms—bidirectional positive interspecies interactions—are abundant and important[1,2]. Traditionally, studies of mutualism have focused on interactions between two species. However, communities often contain many species of mutualists that interact in complex networks[3,4]. For example, many flowering plants are pollinated by multiple insect species[5], and corals interact with a phylogenetically diverse set of endosymbionts[6]. In fact, two-partner mutualisms are now thought to be the exception rather than the norm[3,7]. Understanding the ecology of multiple mutualist communities is an important goal[3,8]. To do so, we need theoretical predictions and experimentally tractable multiple mutualist communities.

In many multiple mutualist systems, several functionally similar species within a "partner guild" supply resources or services to a "shared mutualist," which supplies resources in return (Fig. 1)[3,8]. For example, a guild consisting of two plant species may provide carbon compounds to a shared arbuscular mycorrhizal fungi mutualist that provides both phosphorus, from which one plant species benefits, and pathogen protection, from which the other plant benefits[8,9]. Recent studies have suggested that interactions between species within the partner guild can affect the coexistence and stability of the whole community[10–12]. Within-guild interactions may be especially important if the partner mutualists within the guild are ecologically similar. If partner species' resource niches overlap, they may compete for resources that are external to the mutualism. When multiple species compete for the same limiting resource, one species may competitively exclude the others, leading to a loss of diversity within the community[12,13]. However, because species-rich communities of multiple mutualists exist in nature, certain mechanisms that maintain coexistence must exist[14]. Here, we seek to understand the conditions in which multiple partner mutualists are able to coexist despite competition for a common resource.

Using a community of mutualistic bacteria, we explore the potential for the coexistence of multiple partner species. Our system consists of a partner guild of *Escherichia coli* strains that compete with one another for a carbon source and engage in mutualism with a strain of *Salmonella enterica*, the shared mutualist. The strains engage in mutualism via cross-feeding,

with the *E. coli* strains providing acetate and receiving amino acids from *S. enterica*. We show that the competing *E. coli* strains are unable to coexist when they are provided amino acids in the growth media rather than obtaining them from the shared mutualist because one strain has a faster growth rate. However, when the two *E. coli* strains obtain their amino acids from the shared mutualist, the two *E. coli* strains coexist, maintaining the diversity of the multiple mutualist communities. Next, we use a resource-explicit ecological model to identify factors that promote coexistence. We show that limitation by the shared mutualist is key—if the shared mutualist sets the growth rate of the community, the two-partner mutualists coexist because they are temporarily limited by different resources. Finally, we demonstrate computationally that this phenomenon can promote the coexistence of more than two-partner mutualists. This work helps us understand how diversity is maintained in multiple mutualist communities and can inform efforts to design stable microbial communities.

## Results

**Laboratory experiments.** We studied competition between two-partner mutualists using a laboratory system of cross-feeding bacteria (Fig. 2a). The partner guild consists of one *E. coli* strain that is a methionine auxotroph ("Em") and another *E. coli* strain that is an arginine auxotroph ("Er")—each strain lacks a gene in the biosynthetic pathway for its respective amino acid, so in order for a strain to grow, its required amino acid must be available in the environment. The two *E. coli* strains compete for lactose, which we provide in the growth media, and excrete acetate as a byproduct of lactose metabolism. We experimentally evolved a "shared mutualist" strain of *Salmonella enterica* ("Smr") that secretes methionine and arginine. Smr was derived from a strain that we had previously evolved to secrete methionine[15], and acquired a mutation in *argG* causing arginine secretion. Smr consumes acetate and is unable to metabolize lactose.

Growth rates are a classic measure of competitive ability in microbial systems[16]. Therefore, we started by measuring the growth rates of the three strains in monoculture and in pairwise coculture. For all experiments, we used a batch culture setup in which populations grow until resources are depleted. When grown in monoculture in media containing each strain's required nutrients (with both amino acids in excess in Em and Er monocultures), the three strains have different maximum growth rates (one-way ANOVA: $F(2, 9) = 9897$, $P = 9.1e-16$; Fig. 2b). Er has a slightly higher growth rate than Em (Em: $n = 3$, mean = $0.666\,\mathrm{h}^{-1}$, SD = 0.003; Er: $n = 3$, mean = $0.680\,\mathrm{h}^{-1}$, SD = 0.007; Tukey HSD: $P = 0.037$), and both *E. coli* strains grow faster than Smr (Smr: $n = 6$, mean = $0.172\,\mathrm{h}^{-1}$, SD = 0.007; Tukey HSD: $P < 1e-7$ for both comparisons). When each *E. coli* strain is grown separately in coculture with Smr in lactose media with no amino acids, the coculture containing Em grows faster (Em + Smr: $n = 3$, mean = $0.350\,\mathrm{h}^{-1}$, SD = 0.0006; Er+Smr: $n = 3$, mean = $0.295\,\mathrm{h}^{-1}$, SD = 0.006; Welch's two-sided two-sample $t$ test: $t(2) = 15$, $P = 0.004$; Fig. 2c). This may be because Smr secretes methionine at a faster rate than arginine, or because Em requires less methionine than Er requires arginine (Supplementary Fig. 1). Yields in monoculture and coculture are shown in Supplementary Figs. 2 and 3.

Next, we tested whether the two *E. coli* strains coexist in a lactose environment with excess amino acids and no Smr present. We predicted that the strain with the faster monoculture growth rate, Er, would outcompete Em (Fig. 2b). We assessed the coexistence of the two *E. coli* strains through a mutual invasibility test, measuring whether each *E. coli* strain could increase in frequency when initially rare. Mutual invasibility would indicate

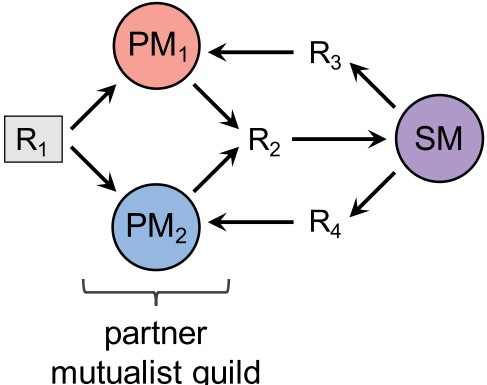

**Fig. 1 A multiple mutualist systems with resource competition between partner species.** The partner mutualist guild is composed of two-partner mutualists ($PM_1$ and $PM_2$) that compete for access to resource $R_1$ (shaded), and produce resource $R_2$. The shared mutualist (SM) consumes $R_2$ and produces two different resources, $R_3$ and $R_4$. $R_3$ is consumed by $PM_1$, and $R_4$ is consumed by $PM_2$. An example of such a community is a shared mutualist arbuscular mycorrhizal fungal species (SM) that provides pathogen protection ($R_3$) to one plant species ($PM_1$) and phosphorus ($R_4$) to another plant species ($PM_2$). The plant species provide carbon ($R_2$) to the shared mutualists and compete for water ($R_1$), which is external to the mutualism.

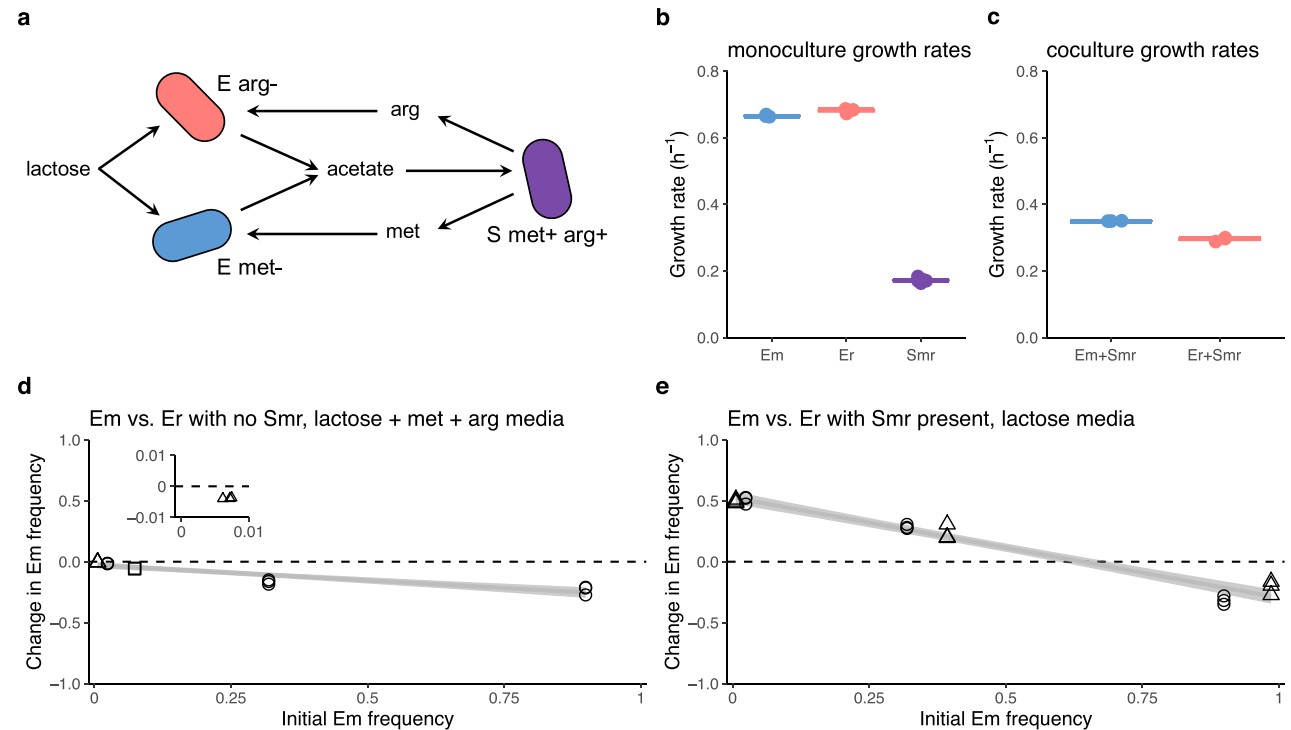

**Fig. 2 Laboratory results show the coexistence of competing partner strains in a multiple mutualist community. a** Schematic showing the interactions between the *E. coli* partner mutualists that comprise the partner guild and the *S. enterica* shared mutualist. The arginine auxotroph, E arg− ("Er"), and the methionine auxotroph, E met− ("Em"), consume lactose and produce acetate. S met+ arg+ ("Smr") consumes acetate and produces both arginine (arg) and methionine (met). **b** The growth rates in monoculture of the three strains differ (one-way ANOVA: $F_{(2, 9)} = 9897$, $P = 9.1e-16$). Er grows faster than Em (Tukey HSD: $P = 0.037$), and Smr grows much more slowly than either *E. coli* strain (Tukey HSD: $P < 1e-7$). Em ($n = 3$) and Er ($n = 3$) were grown in lactose media with excess methionine and arginine, and Smr ($n = 6$) was grown in acetate media. All data points are shown, and the horizontal lines show means. Replicates were independent biological replicates performed over one experiment. **c** Growth rates of Em+Smr ($n = 3$) and Er+Smr ($n = 3$) cocultures. Em+Smr cultures grow faster (Welch's two-sample *t* test: $t(2) = 15$, $P = 0.004$). All data points are shown, and the horizontal line shows the mean. Replicates were independent biological replicates performed over one experiment. **d** A mutual invasibility experiment with cocultures of Em and Er in a lactose medium with excess amino acids. The frequency of Em decreases from all starting frequencies, including when started at 0.6% of the population (inset plot; Welch's two-sample *t* test: $t(2) = 48$, $P = 0.0004$), indicating that Em is the weaker competitor for lactose. The change in Em frequency was calculated as $[Em/(Em + Er)]_{final} − [Em/(Em + Er)]_{initial}$. Supplementary Fig. 4 shows that Em frequency also decreases when Smr is present in this environment. **e** A mutual invasibility experiment in the three-strain multiple mutualist community in a lactose medium. Em increases in frequency when started rare (Welch's two-sample *t* test: $t(2) = −58$, $P = 0.0003$), but decreases in frequency when started common (Welch's two-sample *t* test: $t(2) = 6$, $P = 0.02$), indicating that the two *E. coli* strains coexist. The change in Em frequency was calculated as $[Em/(Em + Er)]_{final} − [Em/(Em + Er)]_{initial}$. Smr yields are similar across all Em frequencies (Supplementary Fig. 7). In panels **d** and **e**, shapes indicate independent experimental batches (A = circles, B = triangles, C = squares). Sample sizes, means, and standard deviations are given in "Results". Source data are provided as a Source Data file.

negative-frequency dependence and coexistence[17,18]. Em decreased in frequency from all five initial frequencies, indicating that Er outcompetes Em and coexistence is not possible (Fig. 2d; Welch's two-sided two-sample *t* test for the lowest initial Em frequency, 0.006: $n = 3$, mean change $= −0.004$, SD $= 0.0001$, $t(2) = 48$, $P = 0.0004$). Competitive exclusion of Em by Er also occurs in this environment in the presence of Smr (Supplementary Fig. 4) and over multiple transfers (Supplementary Fig. 5a).

These findings led to two alternative hypotheses about coexistence in the three-strain multiple mutualist community: *Hypothesis (1)*: One *E. coli* strain will outcompete the other. Er may outcompete Em due to its faster monoculture growth rate and greater competitive ability, or Em may outcompete Er due to its faster coculture growth rate when paired with Smr. *Hypothesis (2)*: The two *E. coli* strains coexist.

To assess the coexistence of the two *E. coli* strains in the three-strain community, we again conducted a mutual invasibility test. We inoculated cultures with six different initial frequencies of the *E. coli* strains, with a constant initial density of Smr. In line with Hypothesis 2, both *E. coli* strains increased in frequency when

initially rare, indicating coexistence (Fig. 2e). Em increased in frequency when initially rare (Welch's two-sided two-sample *t* test for the lowest initial Em frequency, 0.006: $n = 3$, mean change $= +0.495$, SD $= 0.015$, $t(2) = −58$, $P = 0.0003$), and decreased in frequency when initially common (Welch's two-sided two-sample *t* test for the highest initial Em frequency, 0.985: $n = 3$, mean change $= −0.210$, SD $= 0.056$, $t(2) = 6$, $P = 0.02$). Smr also increased in frequency from rare and decreased when initially common (Supplementary Fig. 6), consistent with previous results showing that *S. enterica* stably coexists with cross-feeding *E. coli*[19]. Coexistence was maintained over multiple transfers, although the evolution of prototrophy prevented long-term assessment of frequencies (Supplementary Fig. 5b). Yields are shown in Supplementary Fig. 7.

**Ecological modeling**. To understand why the two *E. coli* strains coexist, we constructed a computational ecological model with ordinary differential equations for the three strains (Em, Er, and Smr) and the four resources (lactose, acetate, methionine, and

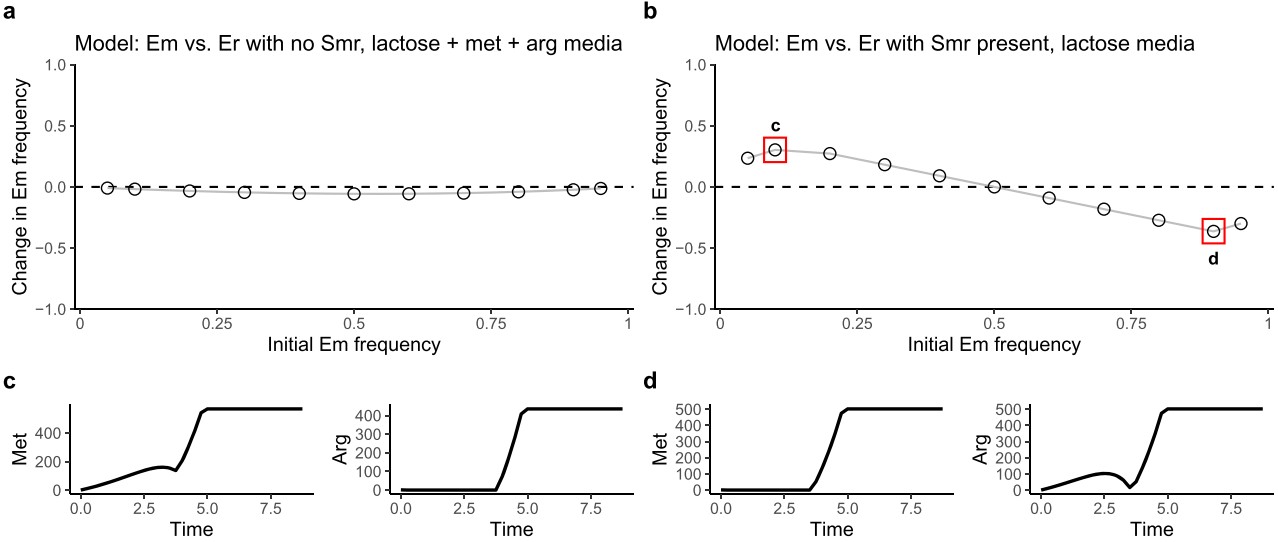

**Fig. 3 A resource-explicit ecological model shows that temporary limitation by different resources promotes coexistence. a** In an Em+Er coculture in an environment with lactose, methionine, and arginine, Em decreases across a range of initial frequencies, indicating that Er is the stronger competitor and that the two strains cannot coexist. **b** In a community of Em, Er and Smr in a lactose environment, both Em and Er are able to increase in frequency when initially rare, which indicates coexistence through negative-frequency dependence. **c** Dynamics of methionine and arginine at the left data point boxed in red in part **b**, where Em begins at 10%. During early timepoints, methionine is not limiting, while arginine is limiting. Growth ceases when lactose is depleted, and dynamics of the three strains and all resources are shown in Supplementary Fig. 9. **d** Dynamics of methionine and arginine at the right data point boxed in red in part **b**, where Em begins at 90%. During early timepoints, arginine is not limiting, while methionine is limiting. The dynamics of strains and all resources are shown in Supplementary Fig. 10. Source data are provided as a Source Data file.

arginine):

$$\frac{d\,Em}{dt} = Em \times \mu_{Em} \times \frac{lcts}{lcts + k_{Em\,lcts}} \times \frac{met}{met + k_{Em\,met}} \quad (1)$$

$$\frac{d\,Er}{dt} = Er \times \mu_{Er} \times \frac{lcts}{lcts + k_{Er\,lcts}} \times \frac{arg}{arg + k_{Er\,arg}} \quad (2)$$

$$\frac{d\,Smr}{dt} = Smr \times \mu_{Smr} \times \frac{ac}{ac + k_{Smr\,ac}} \quad (3)$$

$$\frac{d\,lcts}{dt} = -\frac{dEm}{dt} - \frac{dEr}{dt} \quad (4)$$

$$\frac{d\,ac}{dt} = \left(p_{Em\,ac} \times \frac{dEm}{dt}\right) + \left(p_{Er\,ac} \times \frac{dEr}{dt}\right) - \frac{dSmr}{dt} \quad (5)$$

$$\frac{d\,met}{dt} = -\frac{dEm}{dt} + \left(p_{Smr\,met} \times \frac{dSmr}{dt}\right) \quad (6)$$

$$\frac{d\,arg}{dt} = -\frac{dEr}{dt} + \left(p_{Smr\,arg} \times \frac{dSmr}{dt}\right) \quad (7)$$

Em, Er, and Smr are the population densities of each strain (cells/ml). Resources (lcts = lactose, ac = acetate, met = methionine, arg = arginine) are in units of cell-equivalents/ml (the density of cells that a unit of resource can produce). Growth is governed by Monod saturation rates using Monod constants (e.g., $k_{Em\,lcts}$), which are in units of cells/ml. Production terms (e.g., $p_{Em\,ac}$) are in units of cells/cell. Default values and parameter descriptions can be found in Supplementary Table 1. Briefly, we kept the model simple by using equal values for the same parameters for each of the three strains, except for their growth rates, which we approximated based on monoculture growth rates in the laboratory system (Fig. 2b). Default growth rates are $\mu_{Em} = 1.0$, $\mu_{Er} = 1.1$, and $\mu_{Smr} = 0.5$ with units of 1/timestep.

Consistent with our laboratory system, the model shows that Er outcompetes Em when the two strains are grown in a lactose environment without Smr and with excess amino acids (Fig. 3a). This is because Er has a faster growth rate than Em. Also consistent with our findings in the laboratory system, in the three-strain community in a lactose environment, the two *E. coli* strains coexist. Both *E. coli* strains increase in frequency when started rare (Fig. 3b). Stable equilibrium frequencies are quickly reached when communities are serially transferred with 100-fold dilutions, and coexistence is robust to changes in initial Smr densities (Supplementary Fig. 8).

To understand why Em and Er coexist despite lactose competition, we focused on nutrient dynamics in the model. At the end of growth, lactose is fully depleted while amino acids are in excess, highlighting that competition for lactose sets the yield of each *E. coli* strain (Supplementary Figs. 10 and 11). However, the dynamics of amino acid concentrations early in growth provide an explanation for coexistence. When Em starts rare (the left boxed point in Fig. 3b), it initially depletes little methionine and methionine is therefore plentiful, which allows Em to grow at its maximum growth rate. In contrast, Er initially depletes arginine at a faster rate, and Er's growth rate is limited by low arginine concentrations at early timepoints (Fig. 3c and Supplementary Fig. 9). Conversely, when Er starts rare (the right boxed point in Fig. 3b), arginine is never limiting, while methionine is limiting at early timepoints (Fig. 3d and Supplementary Fig. 10). This means that the initially common *E. coli* strain's growth rate is limited by its amino acid, while the initially rare *E. coli* strain is able to grow at its maximum growth rate because its amino acid is abundant. Initial rates of amino acid consumption create negative-frequency dependence, allowing the initially rare *E. coli* strain to increase in frequency.

To explore the importance of amino acid limitation for coexistence, we used the model to investigate the influence of Smr's growth rate and amino acid production rates. We hypothesized that these parameters are key for coexistence

**a**

Smr growth rate sweep

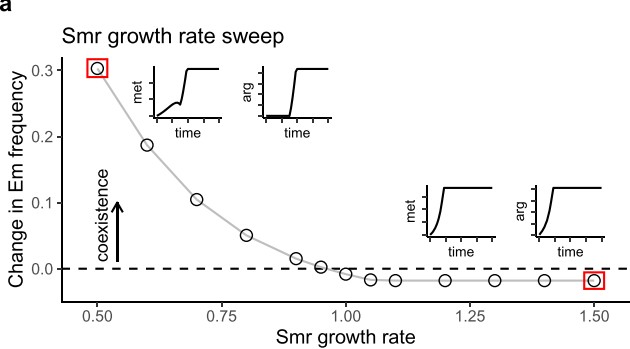

**b**

Smr arg production rate sweep

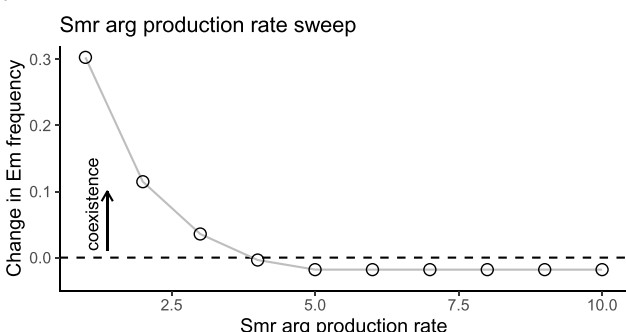

**Fig. 4 Modeling shows that the shared mutualist's growth rate and amino acid production rates affect coexistence. a** Smr growth rate affects coexistence. Growth rates of Em and Er are set at their default levels ($\mu_{Em}$ = 1 and $\mu_{Er}$ = 1.1) and coexistence is evaluated across a range of Smr growth rates. In these simulations, Em begins at a frequency of 0.1, and coexistence is indicated by an increase in frequency. Coexistence is possible when Smr's growth rate is below 0.96. At the left-most point (red box), Em increases in frequency, and the inset plots show that methionine is unlimiting, while arginine is initially limiting. At the right-most point (red box), Em decreases in frequency and the inset plots show that neither methionine nor arginine is limiting at any time point. **b** The rate at which Smr produces arginine also determines coexistence. Em increases in frequency from an initial frequency of 0.1 when the arginine production rate is below 4, but decreases in frequency above this value, indicating that Er takes over the population. Amino acid dynamics at the far left and far right points are similar to the inset plots shown in part **a**. Source data are provided as a Source Data file.

because they affect amino acid limitation. In our laboratory system, Smr grows more slowly than both *E. coli* strains (Fig. 2b), but using our model, we can explore the effect of increasing Smr's growth rate. We increased Smr's growth rate from 0.5 to 1.5. To test for coexistence, we started Em rare (10%) and tracked its change in frequency—an increase in frequency would indicate coexistence, while a decrease would indicate competitive exclusion by the faster-growing Er. When Smr's growth rate is lower than 0.96, Em increases in frequency, and the two *E. coli* strains coexist (Fig. 4a). However, when Smr's growth rate is greater than 0.96, Em decreases in frequency and Er takes over. Under these conditions, methionine and arginine are never limiting (compare Fig. 4a inset plots). This means that both *E. coli* strains grow at their maximum growth rates ($\mu_{Em}$ and $\mu_{Er}$) until lactose is depleted, and the strain with the faster growth rate, Er, takes over ($\mu_{Er} > \mu_{Em}$, see Fig. 2b). Smr's amino acid production rates also affect coexistence. We measured whether Em is able to increase in frequency from rare across a range of arginine production rates, keeping the methionine production rate fixed at 1 and Smr's growth rate at 0.5. When Smr grows more slowly but produces arginine at a rate four times faster than methionine, Em is not

able to invade from rare (Fig. 4b). This is because Er is no longer arginine-limited, and it has a faster maximum growth rate than Em (amino acid dynamics are similar to inset plots in Fig. 4a). In sum, these results suggest that coexistence via amino acid limitation can be created either by a low Smr growth rate or low amino acid production rates.

Another model parameter that we hypothesized could affect amino acid limitation is the rate at which the *E. coli* strains deplete their amino acids (Supplementary Fig. 11). To test for coexistence, we started Em rare (10%) and tracked its change in frequency under different amino acid depletion rates. We found that Er competitively excludes Em when Em consumes methionine rapidly, at nine times the rate at which Er consumes arginine (Supplementary Fig. 11a). Coexistence is lost because methionine becomes limiting for Em, and Er is able to grow faster. Coexistence is also lost when Er's arginine depletion rate is low (around 25% of the default rate; Supplementary Fig. 11b, c). At low arginine depletion rates, both amino acids are abundant throughout growth, and Er is able to grow more quickly and outcompete Em ($\mu_{Er} > \mu_{Em}$, see Fig. 2b). These results support the conclusion that amino acid limitation for the initially common *E. coli* strain is necessary for coexistence. When amino acids are either limiting for both strains or abundant for both strains, coexistence is lost and Er takes over due to its faster growth rate.

Next, we used the model to explore coexistence in a scenario in which the *E. coli* strains deplete both amino acids. In our laboratory system, it is possible that Em depletes arginine and that Er depletes methionine at low rates. We again started Em rare and tracked its change in frequency under different depletion rates of the other *E. coli* strain's required amino acid. We found that depletion of arginine by Em has no effect on coexistence (Supplementary Fig. 12a), presumably because arginine only becomes more limiting for Er at early timepoints. However, if Er depletes methionine at high rates (e.g., at 90% of the rate at which Em depletes methionine), Em is unable to invade and Er takes over due to its faster maximum growth rate (Supplementary Fig. 12b). If both strains are able to deplete the other strain's amino acid, the two effects cancel out and coexistence is always possible, except when both strains deplete the other amino acid at the same per capita rate at which the auxotroph depletes that amino acid (Supplementary Fig. 12c). In that scenario, Er takes over due to its faster maximum growth rate.

We explored complete overlap in amino acid consumption by creating a model in which both *E. coli* strains require and consume the same amino acid and *S. enterica* only produces this single amino acid (Supplementary Fig. 13a). In this situation, the two *E. coli* strains compete both for lactose and for the amino acid, and the slower-growing *E. coli* strain is outcompeted by the faster-growing strain, even if *S. enterica*'s growth rate is low (Supplementary Fig. 13b–d). Consistent with previous simulations, we kept the Monod constants for lactose (e.g., $k_{Em\ lcts}$) and amino acid equal for both *E. coli* strains. As shown elsewhere, unequal Monod constants can allow two competitors to coexist when competing for two essential nutrients[20]. However, in a system like ours where competitors are limited by equivalent resource concentrations, coexistence is not possible with the complete overlap in benefits provided by the shared mutualist.

Finally, we wondered whether frequency-dependent amino acid limitation could promote coexistence in more complex communities. We added a third *E. coli* amino acid auxotroph ("Ex," auxotrophic for a hypothetical amino acid "x") into our model and added production of this amino acid by *S. enterica* ("Smrx") (Fig. 5a and Supplementary Table 2). We set Ex's growth rate slightly lower than Em's, and again assessed coexistence by starting each *E. coli* strain rare and tracking whether it could increase in frequency. When Smrx grows more

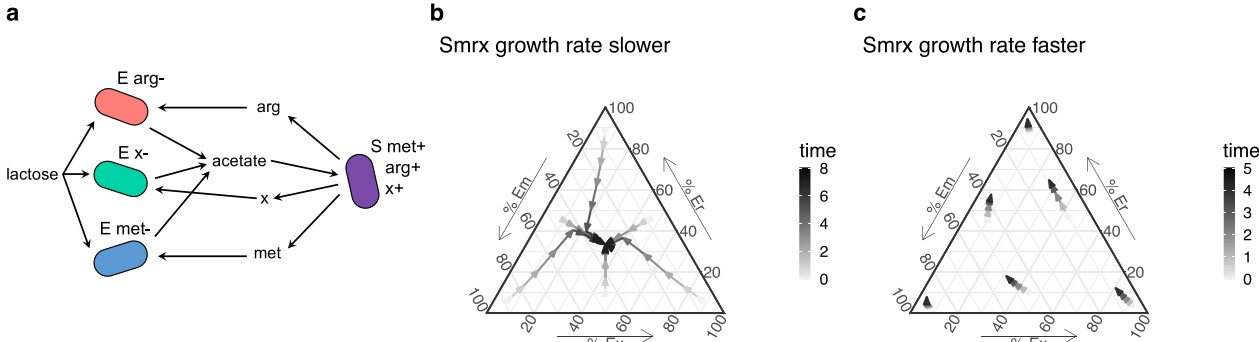

**Fig. 5 Modeling shows that three competing partner mutualists coexist if the shared mutualist sets the community growth rate. a** Schematic showing a community with three *E. coli* partner strains. Ex requires the hypothetical amino acid "x," which Smrx supplies in addition to methionine and arginine, which are consumed by Em and Er, respectively. Equations and parameters are described in Supplementary Table 2. The *E. coli* strain growth rates are $\mu_{Em} = 1$, $\mu_{Er} = 1.1$, and $\mu_{Ex} = 0.9$. **b** A ternary plot showing the frequencies of the three *E. coli* partner strains over time. In these simulations, Smrx's growth rate is 0.5, lower than all three *E. coli* growth rates, and 10,000 units of lactose were supplied so that larger changes in frequencies could be seen within one growth period. All strains are able to increase in frequency when initially rare, indicating coexistence, because the initially rare strains' amino acids are abundant (Supplementary Fig. 14). **c** When Smrx's growth rate is 1.5, the frequency of Er, the faster-growing strain, increases from all starting frequencies, indicating that Er would take over the population over several growth cycles. 10,000 units of lactose were supplied to show larger changes in frequencies. Growth ceases by timepoint 5, so later time points are not shown. Source data are provided as a Source Data file.

slowly than the *E. coli* strains, all three *E. coli* strains coexist (Fig. 5b). However, when Smrx grows faster, the *E. coli* strain with the highest growth rate, Er, outcompetes the other *E. coli* strains (Fig. 5c). The mechanism of coexistence is the same as above, where the amino acid consumed by the initially rare *E. coli* strain (s) is abundant, allowing it to grow at its maximum growth rate, while the initially common strain(s) is limited by its amino acid (Supplementary Fig. 14).

## Discussion
In communities of multiple mutualists, competition between species within a partner guild can affect the coexistence and maintenance of diversity[11,12]. We explored the impact of resource competition between partner species that interact with a shared mutualist on coexistence and stability. Laboratory results showed that two *E. coli* partner mutualist strains that receive different amino acids from a *S. enterica* shared mutualist can coexist, despite the fact that one *E. coli* strain is a better competitor for lactose, the resource that ultimately limits growth. Modeling indicated that stability is possible when the *E. coli* strains are temporarily limited by different resources. While lactose sets the total *E. coli* carrying capacity—growth ceases when lactose is exhausted—the availability of amino acids during growth determines the instantaneous growth rate of each *E. coli* strain. When one *E. coli* strain begins rare, its amino acid is always abundant, so its instantaneous growth rate is faster and it consumes lactose quickly. In contrast, the initially common *E. coli* strain is amino acid-limited at early timepoints and grows more slowly than the rare strain, which allows the initially rare *E. coli* strain to increase in frequency. The community is therefore stable through negative-frequency dependence. We found that three key parameters affect the potential for coexistence via temporary amino acid limitation. Coexistence is not possible if *S. enterica*'s growth rate is high, if *S. enterica*'s production rate of the initially common *E. coli*'s amino acid is high, or if the initially common *E. coli* strain depletes its amino acid at a low rate. In these situations, the initially common *E. coli* strain is never limited by its amino acid, and the stronger competitor excludes the weaker. In summary, coexistence requires temporary amino acid limitation for one partner strain.

This mechanism of stability is related to classical ideas in ecology about niche partitioning[21,22]. Theoretical work predicts

that multiple species typically do not coexist if they are limited by the same resource[13,20] (though see Lobry et al.[14]). However, if the species are limited by different resources, they can more easily coexist[23,24]. In our system, the carrying capacity of the *E. coli* strains is ultimately limited by lactose. However, during growth, the limiting resources are temporarily "partitioned." One strain's instantaneous growth rate is limited by its amino acid, while the other achieves its maximum growth rate due to an abundance of lactose and its amino acid. An interesting element of our system is that a biotic factor creates the potential for temporary niche partitioning, rather than an aspect of the environment. The shared mutualist, *S. enterica*, causes the two-partner species' instantaneous growth rates to be determined by different resources early on in growth. In addition, the shared mutualist creates the potential for niche partitioning by providing two different resources for the partner strains. Coexistence is not possible if both partner strains receive the same resource from the shared mutualist (Supplementary Fig. 13).

Recent work has explored the importance of competition between partner species in multiple mutualist communities. Several empirical studies have documented competition between species within a partner guild for access to the shared mutualist. For example, flowering plants compete for pollinator services[25], and multiple species of plant-defending ants compete for nesting sites on host acacia plants[26]. In these cases, partner species may also compete for resources that are external to the mutualism. For example, in the ant–plant systems, plants may compete with one another for water and nutrients, and ants for prey[12]. Johnson and Bronstein[12] took a mathematical approach to examine the coexistence of two-partner mutualists that compete for both a host-provided resource and an external resource. They determined that coexistence requires that one partner is limited by the host-provided resource and the other by the external resource (i.e., niche partitioning). Together with this study, our results suggest that understanding the stability of multiple mutualist systems requires consideration of competition for external resources in addition to competition for access to the shared mutualist. We show that even when both competing partners are ultimately limited by a resource external to the mutualism, coexistence can be maintained through temporary niche partitioning. Our work also identifies the importance of the shared mutualist providing different resources to members of the partner mutualist guild.

Microbial communities are often observed to include many cross-feeding species that exchange metabolites[27–29]. An open question in microbial ecology is why natural communities appear to contain several ecologically similar species that consume the same resources and carry out the same functions[29,30]. Our work suggests that these communities may be stable despite the potential for competition between strains that provide redundant functions (in our case, the conversion of lactose to acetate). We also showed that temporary limitation by different resources allows for the coexistence of three partner strains (Fig. 5b), and this mechanism may extend to the coexistence of many partner strains. However, there is likely a limit to the number of metabolites that a single shared mutualist can secrete and therefore an upper limit to the system complexity. Other factors that are likely to influence the stability of cross-feeding systems include spatial structure and evolution. In general, the spatial structure promotes diversity[31], though structure can also lead to a loss of strains[32]. Evolution can lead to rapid changes in cross-feeding[33,34]. The evolution of specialists that only interact with a subset of competing partners may decrease the diversity of the system. This will be explored in future work. Finally, our mechanism of coexistence relies on the dynamics created by a batch or seasonal culture regime. However, analytical analysis of a chemostat model of our system indicates that coexistence is also possible in continuous culture, through co-limitation by both amino acids and lactose (Supplementary Note 1).

The results presented here improve our understanding of the ecology of multiple mutualist communities, expanding our knowledge of mutualisms beyond pairwise interactions. Ecological stability is critical for the maintenance of biodiversity. Within mutualistic communities, the coexistence of many species within a partner mutualist guild creates functional redundancy, which is important in the face of disturbances because redundancy can protect mutualistic communities from collapse[5,35]. Knowledge of ways to preserve functional redundancy can aid efforts to design stable microbial communities for applications in health and industry[36].

## Methods

**Strains and media**. We used two *Escherichia coli* K12 strains, both derived from the Keio collection[37]. The methionine auxotroph ("Em") has a $\Delta metB$ mutation, and the arginine auxotroph ("Er") has a $\Delta argA$ mutation. *LacZ* was added to both strains using conjugation for Em, and phage transduction for Er[38]. Em has a cyan fluorescent protein inserted in the lambda attachment site. We also used a *Salmonella enterica* serovar Typhimurium LT2 strain that secretes methionine and arginine ("Smr"). This strain was derived from a strain containing mutations in *metA* and *metJ* that cause overproduction of methionine[15,39,40]. We selected for arginine production by coculturing this strain with the *E. coli* arginine auxotroph as a lawn on lactose minimal media plates containing x-gal (0.05% v/v) for four 7-day growth cycles with 1:6.67 dilutions at each transfer[41]. The appearance of a blue colony suggested the evolution of arginine production in *S. enterica*, which we confirmed by isolating *S. enterica* from the colony on citrate minimal media plates and cross-streaking with Er. We sequenced this strain using Illumina NextSeq and identified mutations using breseq[42]. We found a T → A point mutation in *argG* at position 3459818 (reference strain NC_003197).

In coculture or three-strain cultures, strains were grown in a modified Hypho minimal medium with lactose as the carbon source, containing 2.78 mM lactose, 14.5 mM $K_2HPO_4$, 16.3 mM $NaH_2PO_4$, 0.814 mM $MgSO_4$, 3.78 mM $Na_2SO_4$, 3.78 mM $[NH_4]_2SO_4$, and trace metals (1.2 μM $ZnSO_4$, 1 μM $MnCl_2$, 18 μM $FeSO_4$, 2 μM $(NH_4)_6Mo_7O_{24}$, 1 uM $CuSO_4$, 2 mM $CoCl_2$, 0.33 μm $Na_2WO_4$, 20 μM $CaCl_2$). Monocultures and cocultures of Em and Er were grown in this medium with 250 μM of methionine and 250 μM of arginine added, concentrations that we found to be unlimiting (i.e., growth ceased when lactose was depleted, rather than the amino acids; Supplementary Fig. 1). Smr's growth rate in monoculture was assessed in Hypho minimal medium with 12 mM acetate rather than lactose, a concentration that approximates the total amount of acetate produced by the *E. coli* strains.

To measure final yields as colony-forming units (CFU/ml), cultures were diluted in saline (0.85% NaCl) and plated on selective Hypho minimal media plates with 1% agar. Plates for the *E. coli* strains contained 2.78 mM lactose and 100 μM of methionine for Em, or 100 μM of arginine for Er. Smr was plated on Hypho plates containing 3.4 mM sodium citrate instead of lactose. All plates contained

0.05% v/v x-gal, which makes *E. coli* colonies blue. We counted the dilution plates that had between 30 and 300 colonies.

**Growth assays**. All experiments were performed in 96-well plates with 200 μl of media per well, inoculated with a 1:200 dilution of log-phase monocultures (1 μl of each strain). We measured $OD_{600}$ in a Tecan InfinitePro 200 plate reader at 30 °C, shaking at 432 rpm between readings, which were taken every 20 min. Growth rate estimates were calculated by fitting growth curves to a Baranyi function[43] by obtaining nonlinear least-square estimates and using the growth rate parameter estimate. For all experiments (monocultures, cocultures, and mutual invasibility experiments), we performed three biological replicates, except for Smr monocultures, where we performed six biological replicates. All measurements were taken from distinct samples.

**Mutual invasibility experiments**. The ability of each *E. coli* strain to increase in frequency from rare was our criterion for coexistence[17,18]. To measure whether invasion from rare occurred, we set up cultures with different starting frequencies and tracked their changes over one growth cycle. We performed several batches of these experiments on different weeks, with three biological replicates per batch. The initial Smr density was kept constant (9.8e5 CFU/ml for batch A, 3.7e5 CFU/ml for batch B), and the *E. coli* total density was kept constant (1.6e6 CFU/ml ± 5.4e5 (SD) for batch A,_1.7e6 CFU/ml ± 5.3e5 (SD) for batch B, 3.6e6 ± 6.7e5 (SD) for batch C) but the frequency of each strain differed across a range of frequencies —0.024, 0.320, and 0.899 Em/(Em + Er) for batch A, 0.006, 0.393, and 0.985 Em/(Em + Er) for batch B, and 0.08 for batch C. (Note that batch C was only performed in lactose + methionine + arginine media with no Smr present.) After growth, the cultures were diluted and plated on strain-specific plates to measure yields as CFU/ml (media described above). The change in Em frequency was calculated as $[Em/(Em + Er)]_{final} - [Em/(Er + Er)]_{initial}$.

**Ecological modeling**. The ecological model is shown in "Results" and Supplementary Table 1. The ODE system was solved using the deSolve package in R, which used the lsoda solver to numerically integrate. All simulations were solved for sufficient duration to ensure dynamics had ceased. During integration, relative tolerance (rtol) was set to 1e-13 and maxsteps to 1e5.

**Analysis and statistics**. Modeling, data visualization, and statistical analyses were done in R version 3.6.0. Monoculture growth rates were compared using a one-way analysis of variance (ANOVA) and post hoc Tukey HSD tests, with $\alpha = 0.05$. The coculture growth rates and mutual invasibility experiments were analyzed using Welch's two-sided two-sample *t* test, which is designed for unequal sample distribution variance and assumes normality, with $\alpha = 0.05$. We used the R package ggtern to make the ternary plots in Fig. 5.

**Reporting summary**. Further information on research design is available in the Nature Research Reporting Summary linked to this article.

## Data availability
All data generated and/or analyzed in this study are provided in the Source Data file and are also available in the Zenodo repository at https://doi.org/10.5281/zenodo.432179[44]. Source data are provided with this paper.

## Code availability
Code for the ecological model is available in the Zenodo repository at https://doi.org/10.5281/zenodo.432179[44]. Code for data analysis is available upon reasonable request.

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

## Acknowledgements

We thank Lisa Fazzino, Ben Holte, and Alejandro Behling for assistance with preliminary experiments, Jeremy Chacón for help with modeling, and Brian Smith, Leno Smith, and Jonathan Martinson for providing comments on the manuscript. This work was supported by the National Institutes of Health (1R01-GM121498 to W.R.H. and U01EB019416 to R.P.C.), and by a National Science Foundation Graduate Research Fellowship for S.P.H.

## Author contributions
S.P.H. conceived the study, collected and analyzed the data, and prepared the paper. W. H. helped design the experiments, oversaw data analysis, supplied materials, and assisted with the paper preparation. T.G. analyzed the chemostat model and provided comments on the paper. R.C. constructed the Er strain, provided ideas for additional modeling experiments, and provided comments on the paper.

## Competing interests
The authors declare no competing interests.
