## [Peer Review File · Nature Communications]

Reviewers' Comments:

Reviewer #1:

Remarks to the Author:

This manuscript describes an interesting study of how mutualistic relationships can increase ecological diversity. The authors show that two strains of *E. coli*, which each lack a gene in the biosynthetic pathway for two different amino acids, are able to grow in a lactose-limiting medium in which the two amino acids are supplied in excess, but the authors argue (but see below) that they cannot coexist in this medium. Instead, they can coexist in the lactose-limiting medium without the amino acids and in the presence of a mutualist partner species, *Salmonella enterica*, which secretes the two amino acids and consumes acetate, the byproduct of lactose consumption by the *E. coli* strains. The experimental system is easy to understand and the results are enlightening. Additionally, the use of a simple model proposes a mechanism for the coexistence—negative frequency-dependence, mediated by differences in amino acid concentration depending on which strain of *E. coli* is more frequent. Overall, I felt that the study has significant potential, and would be appropriate for publication in *Nature Comm* if the authors can address the points below.

My primary concern has to do with whether the authors have adequately demonstrated that the two *E. coli* strains cannot coexist in the absence of their mutualist partner. Fig 2d show the relevant co-culture experiments, where the conclusion regarding competitive exclusion rests on the where the authors started with 2% of *Em*.

1. How many replicates were done for this condition? In the figure there was only one visible data point, but I assume that there were more cultures that were done?
2. How many colonies were counted for each data point shown? Note that Poisson counting gives a minimal error for each measurement. It is unlikely that there can be a statistically significant effect given the numbers of colonies that have to be counted (and the potential transient effects discussed below).
3. For an experiment like this I would feel more comfortable knowing that the entire experiment was repeated on a separate week, as there can be batch effects.
4. More generally, in my group we have found that these invasion assays are much more reliable if done over multiple growth-dilution cycles. This is because the measured change in fractions during the first day can depend in a large manner depending upon the conditions of the cultures that the cells are taken from (eg time of saturation etc). In particular, there can be transient invasion for a variety of reasons.
5. Moreover, if the authors did multiple growth-dilution cycles then they could measure the equilibrium properties of all of the communities (fractions, population size, etc).

More generally, I was hoping to see some of the predictions from the model confirmed in the experiments. I am convinced regarding the core mechanism regarding how the mutualism interaction stabilizes the coexistence of the *E. coli* species, but the paper would have felt more rich and complete if there were more of an interplay between the theory and experiment. This is especially true because the core phenomenon is a bit simple: Two amino acid auxotrophs will always coexist when amino acids are limiting (such as in the three species community of Fig 2e), and coexistence is not expected when amino acids are in excess (an important point that is not at first obvious in the paper is that Fig 2d and 2e are different not only in the presence/absence of *Smr* but also in whether the amino acids are supplied in excess). Supplementing the three-species community with excess amino acids would provide further evidence for the authors' model, if the two strains of *E. coli* could not coexist there.

I found the discussion of the continuous chemostat situation to be a bit odd. In particular, I don't understand why the mechanism of coexistence is qualitatively different, since it seems to me to be something that I would expect and for the same reason that it happens in batch culture (note that batch culture with frequent dilutions is a chemostat). The analysis of the chemostat in the supplement is very mathematical but does not (obviously) include any intuitive discussion of how the chemostat situation is different from the batch situation.

Better explanation of the simulation results would also be useful. In particular:

- The paragraph beginning on line 153 first describes the negative frequency dependence of the model, but indirectly. I wonder if the authors are trying to introduce the concept slowly in order to make it less confusing, but I found it more confusing. By only describing the abundance of the amino acids over time, they leave out the fact that the abundance of each of the strains is causing these dynamics, making this point appear to be mysterious simulation result.
- It is not clear enough that the results in the paragraph beginning on line 163 are due to E_r 's higher monoculture growth rate. This paragraph should explain this, explicitly clarify that "the strain with the faster growth rate" (line 173-174) is E_r (and refer to Fig. 2b), and explicitly define maximum growth rate as μ in the model.
- In the paragraph beginning on line 182, it is again not clear that growth rate is the reason for these simulation results. The description makes it seem like the model has an asymmetry for no apparent reason, and a more nuanced discussion is warranted to explain why changes in amino acid depletion change the outcomes.
- Also about this paragraph: the section about the version of the model with only one amino acid consumed by both *E. coli* strains would be better as a separate paragraph. And it should be explained why three species cannot coexist in a three-resource environment, given the right parameters. The text describes how changing the growth rates does not allow coexistence, but does not describe changing the Monod constants—why won't trade-offs between growth rates and Monod constants allow for coexistence of three species in three resources?
- In the paragraph beginning on line 202, is there any reason to name the third amino acid in the simulation? Mentioning phenylalanine implies that it has a unique property, or even that you did another experiment with a third strain. If there is nothing special about phenylalanine, it should be renamed as a generic third amino acid to avoid confusion.

Finally, the flow of the manuscript (and particularly the Results section) is not ideal. The transition from experiments to simulations is not clear, and the reader might think that simulation descriptions are referring to experiments (the simulation figures should also be clearly labeled, either in the figure title or the image).

Minor issues:

- ✓ In some cases, adding a final sentence to the paragraphs in the Results section that sums up the findings (and/or say what they mean) could make the text more clear.
- ✓ Fig S4 is hard to interpret, which makes the argument for coexistence in the 3-species community less persuasive. Maybe adding initial abundances to the plot would help.

Jeff Gore

Reviewer #2:

Remarks to the Author:

Review of "Limitation by a shared mutualist promotes coexistence of multiple competing partners" by Hammarlund et al.

In their manuscript, the authors explore to what extent a shared mutualist partner facilitates the coexistence between two species competing for common resources. The manuscript is made of two parts: an experimental part and a modelling part. I particularly enjoyed seeing the interplay between the experiment and the model parts and, the authors explaining the experimental results based on a mechanistic model. As a theoretician, I cannot assess whether the experiment was performed

carefully, but the design was done properly for answering the questions. My review will focus on the modelling part.

Although I enjoyed reading the manuscript, I found inconsistencies in the theoretical part. It thinks it can be corrected, but this will require a substantial amount of work and literature reading.

My first point is about the "Competitive exclusion principle". This principle has been largely overstated in ecology. This is mainly due that people tend to forget under which assumptions such a principle is valid. Actually, it holds when there is no other interactions between and within species than the "indirect" interaction of competition due to the single common resource. As soon, as there is another source of intraspecific or interspecific interaction (we can simply thin on space limitations or direct encounter among individuals) the principle doesn't hold, and there can be enough niche differentiation to make species coexisting. The authors should consult the work of Claude Lobry, such as "Persistence in ecological models of competition for a single resource" (doi:10.1016/j.crma.2004.12.021) or his two books: chapter 3.6 of "The Consumer-Resource Relationship" and chapter 3 of "The Chemostat"

My second point is about the invasion criterion for coexistence. Again this is a principle that has been largely overstated in ecology and its validity is not as general as it is claimed by several authors. Actually, the invasion criterion is mathematically proven in the case of two competing species described by a Lotka-Volterra model, i.e.

$$\begin{aligned}dN_1 / dt &= N_1 (r_1 - \alpha_{11} N_1 - \alpha_{12} N_2) \\dN_2 / dt &= N_2 (r_2 - \alpha_{21} N_1 - \alpha_{22} N_2)\end{aligned}$$

In this model, if species 1 can invade the monoculture equilibrium of species 2 and vice versa, then both species coexist. Then this criterion has been thought, without justification, to be true whatever the number of species is and the dynamic model is. Recently in Saavedra et al. ("A structural approach for understanding multispecies coexistence", doi: 10.1002/ecm.1263) have shown that coexistence, with 3 species and a Lotka-Volterra model, can be reached without the invasion principle to be fulfilled (figures 6, 7, and 9). Thus, as stated in the manuscript I'm not a priori convinced that the invasion criterion is equivalent to coexistence in the batch model proposed by the authors on page 5 of their manuscript. The authors have to prove that it holds for their specific model, if not it is pure speculation.

At all, I find it is a very interesting study that may deserve publication if the authors adjust and correct their theoretical part.

Reviewer #3:

Remarks to the Author:

First of all, I am sorry this review took me so long. These have been extraordinary circumstances.

This paper explores a classical question in ecology: how can two or more species coexist if they compete for the same resources? In this case, however, these two competing species share a mutualist partner. It is then, the temporal variation created by interactions within this mutualism that allows for coexistence. If two (or more) species obtain different resources from the same partner, then, they can coexist even if they compete for all other resources, as long as the shared partner depends on the mutualists to grow and does not grow too fast. This paper is a very neat example combining experiments and models and has wide implications for thinking on how different kinds of interactions affect community assembly and stability.

This paper proposes a mechanism of coexistence, that to my knowledge has not been applied to multispecies mutualisms. As such, I think this is a paper that inspires multiple avenues of research

(e.g. How common is this mechanism in nature? How evolutionary stable is this mechanism? Under which conditions is it stable?).

I have two larger concerns/questions, but I think they are not hard to address. First, I am not sure that the results from the mutual invasibility experiment hold-up when you have multiple species, and the community is not allowed to reach equilibrium (although from your model I think your conclusions hold-up). The problem is that you could have added initially a low density of Salmonella (such that it exacerbates the temporal distribution of nutrients) but over a few transfers, as the species change in relative abundance, you might increase the density of Salmonella making cross-feeding resources less limiting and reducing the effect of temporal variation. Have you done competition experiments with multiple transfers to evaluate what are the relative frequencies of the three strains at equilibrium? Am I missing something?

Second (and I feel some hesitation about bringing this up given COVID-19) I felt like the experimental observation was a great motivation for the model. However, the model makes some interesting predictions and I was left hoping for some of those to be evaluated experimentally. For example, it might not be possible to change the growth rate of Salmonella, but it would be interesting to know how coexistence changes across a gradient of Salmonella initial density (prior to equilibration), or measure the growth rates of the E. coli strains when grown in the supernatant of Salmonella after different periods of growth in the presence of each of the E.coli strains. These experiments would, in my opinion, make the paper stronger, but they are not necessary.

Finally, I had a couple of smaller comments and questions:

P2-L32: Most bees get both pollen and nectar from flowers. I would check this to avoid making pollinator biologists mad.

P5- Er and Em equations: I am not sure if I completely understood why were the consumption of lactose and methionine or arginine terms multiplied and not added. Is there co-utilization?

P5-L145: I also was not sure why did you use these growth rates instead of the ones you measured?

P6-L183: "We found that the rate at which Em depletes methionine has no effect on coexistence" - That is interesting! Why?

P8-251: What if strains compete for the same resource in the mutualism but different external resource? I think this parallels many mutualisms in nature, and it would be interesting to know (maybe not in this paper?) under which conditions is this possible.

Fig 3a (and S9) Why is there a little dip in the change of Em frequency at intermediate frequencies of Em?

Reviewer #1:

This manuscript describes an interesting study of how mutualistic relationships can increase ecological diversity. The authors show that two strains of E coli, which each lack a gene in the biosynthetic pathway for two different amino acids, are able to grow in a lactose-limiting medium in which the two amino acids are supplied in excess, but the authors argue (but see below) that they cannot coexist in this medium. Instead, they can coexist in the lactose-limiting medium without the amino acids and in the presence of a mutualist partner species, Salmonella enterica, which secretes the two amino acids and consumes acetate, the byproduct of lactose consumption by the E coli strains. The experimental system is easy to understand and the results are enlightening. Additionally, the use of a simple model proposes a mechanism for the coexistence—negative frequency-dependence, mediated by differences in amino acid concentration depending on which strain of E coli is more frequent.

Overall, I felt that the study has significant potential, and would be appropriate for publication in Nature Comm if the authors can address the points below.

*My primary concern has to do with whether the authors have adequately demonstrated that the two *E. coli* strains cannot coexist in the absence of their mutualist partner. Fig 2d show the relevant co-culture experiments, where the conclusion regarding competitive exclusion rests on the where the authors started with 2% of *Em*.*

1. How many replicates were done for this condition? In the figure there was only one visible data point, but I assume that there were more cultures that were done?

We performed three biological replicates for each initial *Em* frequency, which we now state in the figure legend and methods. We also have added an inset plot for the lowest initial *Em* frequency points in Fig. 2d, and jittered the points to show all three replicates.

2. How many colonies were counted for each data point shown? Note that Poisson counting gives a minimal error for each measurement. It is unlikely that there can be a statistically significant effect given the numbers of colonies that have to be counted (and the potential transient effects discussed below).

We followed the common rule-of-thumb of counting between 30 and 300 colonies for each strain (which we've now added to the methods section). Our selective media plates allow for good counts of all strains, even for the strain(s) in the minority. We plated many 10-fold dilutions, and counted the plates with 30-300 colonies.

For example, for the three biological replicates of the lowest initial frequency of *Em* in Fig. 2d, we counted 245, 241, and 196 colonies on lactose + methionine plates that had 100ul of 10e3-fold diluted culture spread onto them. For *Er*, we counted 109, 118, and 99 colonies on 10e6-fold diluted lactose + arginine plates.

Tomasiewicz et al. 1980 discuss the 30-300 colony tradition and the problems associated with low and high colony counts, and recommend counting around 25-250 colonies. In other experiments with this system, we've done replicate platings from the same culture, and found very low variance, so we feel confident about our CFU/ml estimates.

3. For an experiment like this I would feel more comfortable knowing that the entire experiment was repeated on a separate week, as there can be batch effects.

We repeated the experiment shown in Fig. 2d twice more, on separate weeks, for two different initial *Em* frequencies. This is the condition in which we expect to see competitive exclusion of *Em* by *Er*. We tested a lower initial *Em* frequency (0.006) than our original lowest frequency

(0.024). When Em begins at 0.006 ($E_m/(E_m+E_r)$), we observed a -0.004 change in Em frequency (standard deviation +/- 0.0001), which was significantly different from 0 ($t = 47.6$, $df = 2$, $p\text{-value} = 0.0004$). The three batches are represented by different point shapes, and the inset shows the lowest Em frequency.

We also repeated the experiment shown in Fig. 2e. We performed one more batch, with three initial Em frequencies. The initial frequencies for this batch were slightly more extreme, with Em starting at a frequency of 0.006 on the low end, and 0.985 on the high end. Consistent with the original experimental results, Em increased in frequency from the lowest initial frequency (a change of +0.5, s.d. +/- 0.014, $t = -58.1$, $df = 2$, $p\text{-value} = 0.0003$), and decreased in frequency from the highest initial frequency (-0.21, s.d. +/- 0.056, $t = 6.47$, $df = 2$, $p\text{-value} = 0.023$). The two batches are represented by different point shapes.

4. More generally, in my group we have found that these invasion assays are much more reliable if done over multiple growth-dilution cycles. This is because the measured change in fractions during the first day can depend in a large manner depending upon the conditions of the cultures that the cells are taken from (eg time of saturation etc). In particular, there can be transient invasion for a variety of reasons.

We agree that there are benefits of assessing coexistence by doing transfers. In response to this comment, we performed transfer experiments, and the results were consistent with our hypotheses for the first 3-4 transfers. Em frequency rapidly dropped when in competition with Er in an amino acid-replete environment, while Em and Er coexisted in a lactose environment with Smr present. However, we began to see deviations from these trends around transfer 5.

In past experiments with our Er strain, we've seen that it rapidly reverts to prototrophy, presumably by gaining mutations that compensate for the *argA* deletion. We wondered whether this had occurred in our transfer experiments.

We plated cultures from transfer 5 onto lactose minimal media plates with no amino acids, and observed roughly the same number of colonies on those plates as on lactose + methionine and lactose + arginine plates (which we use to count Em and Er, respectively). Our Em and Er strains are unable to form colonies on lactose plates, so this was a sign that reversion to prototrophy had occurred. We grew eight randomly chosen colonies from the lactose plates in our plate reader and observed no CFP fluorescence (our Em strain produces CFP, Er does not), leading us to conclude that these colonies were Er-derived mutants that had reverted to prototrophy and invaded the population.

We have concluded that the rapid evolution in Er makes our system ill-suited for multiple transfer experiments. Though Er prototrophic mutants did not rise to an appreciable frequency in our transfer experiments for several transfers, we have conservatively decided not to present data beyond transfer 1. We have evidence that prototrophic colonies don't arise to a detectable

frequency after one growth period. The data from the first growth cycle of the Em Er in lactose + methionine + arginine treatment are shown as square points in the updated Fig. 2d.

While transfer experiments have the benefit mentioned below of measuring equilibrium abundances, several other studies employ mutual invasibility assays similar to ours. Two example studies that use mutual invasibility assays over one growth cycle to demonstrate negative frequency dependence are Grieg and Travisano 2004 (*The Prisoner's Dilemma and polymorphism in yeast SUC genes*) and Pande et al. 2014 (*Fitness and stability of obligate cross-feeding interactions that emerge upon gene loss in bacteria*).

5. Moreover, if the authors did multiple growth-dilution cycles then they could measure the equilibrium properties of all of the communities (fractions, population size, etc).

We agree that this would be interesting, and regret that the evolutionary lability of Er prevents us from doing this. When performing transfers in the model, we find that population sizes quickly stabilize. The plot below shows the density of Em, Er, and Smr when Em begins at 10%, with 100-fold dilutions when transferring and default parameters.

This plot is one panel (top middle) of a new Supplemental Figure 7 that shows additional transfer experiments in the model that we performed in response to a comment from reviewer 3, described below.

More generally, I was hoping to see some of the predictions from the model confirmed in the experiments. I am convinced regarding the core mechanism regarding how the mutualism interaction stabilizes the coexistence of the E. coli species, but the paper would have felt more rich and complete if there were more of an interplay between the theory and experiment. This is

especially true because the core phenomenon is a bit simple: Two amino acid auxotrophs will always coexist when amino acids are limiting (such as in the three species community of Fig 2e), and coexistence is not expected when amino acids are in excess (an important point that is not at first obvious in the paper is that Fig 2d and 2e are different not only in the presence/absence of Smr but also in whether the amino acids are supplied in excess). Supplementing the three-species community with excess amino acids would provide further evidence for the authors' model, if the two strains of *E. coli* could not coexist there.

Thank you for the suggestion of supplementing the three species community (Em, Er, Smr) in lactose with excess methionine and arginine. We did this alongside the second batch of Em Er competitions in lactose + methionine + arginine media for the lowest Em frequency, and found very similar results. When Em began at a frequency of 0.006, it decreased in frequency both when Smr was absent and present, suggesting that competitive exclusion occurs regardless of Smr's presence or absence. The figure below is now shown in the SI as Supplementary Figure 4.

With regards to the comment about testing predictions from the model in the lab system: We wish we were able to easily manipulate features of the system that we explored in the model, like Smr's growth rate and amino acid production rates. Unfortunately, these experiments would not be trivial to set up, and we are constrained by COVID-19 restrictions on time spent in the lab. For future work, we plan to generate a construct of the Smr strain that would allow us to manipulate amino acid secretion rates. We also plan to do experiments with a three *E. coli*

strain community. We regret that we are unable to incorporate those experiments into this manuscript.

I found the discussion of the continuous chemostat situation to be a bit odd. In particular, I don't understand why the mechanism of coexistence is qualitatively different, since it seems to me to be something that I would expect and for the same reason that it happens in batch culture (note that batch culture with frequent dilutions is a chemostat). The analysis of the chemostat in the supplement is very mathematical but does not (obviously) include any intuitive discussion of how the chemostat situation is different from the batch situation.

Thank you for this comment. We initially posed the mechanisms as distinct because of the temporal aspect of the limitation. In batch culture, *E. coli* growth ends when lactose is depleted. Amino acids modulate early growth rates but are in excess by the end of batch growth. In contrast, in continuous chemostat culture, growth is co-limited by lactose and amino acids. However, we agree that limitation by different amino acids ultimately drives coexistence in both cases, so we have revised the wording in the manuscript to more appropriately reflect this.

Line 332: However, analytical analysis of a chemostat model of our system indicates that coexistence is also possible in continuous culture, through co-limitation by both amino acids and lactose (Supplementary Information Section 2).

We have also edited the chemostat section of the supplementary information to make the discussion more intuitive.

Better explanation of the simulation results would also be useful. In particular:

- The paragraph beginning on line 153 first describes the negative frequency dependence of the model, but indirectly. I wonder if the authors are trying to introduce the concept slowly in order to make it less confusing, but I found it more confusing. By only describing the abundance of the amino acids over time, they leave out the fact that the abundance of each of the strains is causing these dynamics, making this point appear to be mysterious simulation result.*

We expanded this paragraph and now make it clear how each of the strains affect amino acid dynamics:

Line 165: To understand why E_m and E_r coexist despite lactose competition, we focused on nutrient dynamics in the model. At the end of growth, lactose is fully depleted while amino acids are in excess, highlighting that competition for lactose sets the yield of each *E. coli* strain (Supplementary Fig. 8 and 9). However, the dynamics of amino acid concentrations early in growth provide an explanation for coexistence. When E_m starts rare (the left boxed point in Fig. 3b), it initially depletes little methionine and methionine is therefore plentiful, which allows E_m to

grow at its maximum growth rate. In contrast, Er initially depletes arginine at a faster rate, and Er's growth rate is limited by low arginine concentrations at early timepoints (Fig. 3c, Supplementary Fig. 8). Conversely, when Er starts rare (the right boxed point in Fig. 3b), arginine is never limiting, while methionine is limiting at early timepoints (Fig. 3d, Supplementary Fig. 9). This means that the initially-common *E. coli* strain's growth rate is limited by its amino acid, while the initially-rare *E. coli* strain is able to grow at its maximum growth rate because its amino acid is abundant. Initial rates of amino acid consumption create negative-frequency dependence, allowing the initially-rare *E. coli* strain to increase in frequency.

• It is not clear enough that the results in the paragraph beginning on line 163 are due to Er's higher monoculture growth rate. This paragraph should explain this, explicitly clarify that "the strain with the faster growth rate" (line 173-174) is Er (and refer to Fig. 2b), and explicitly define maximum growth rate as μ in the model.

Thanks for this suggestion. We have made these changes. The second half of this paragraph appears below with changes in bold:

Line 190: However, when Smr's growth rate is greater than 0.96, Em decreases in frequency and Er takes over. Under these conditions, methionine and arginine are never limiting (compare Fig. 4a inset plots). This means that both *E. coli* strains grow at their maximum growth rates (**μ_{Em} and μ_{Er}**) until lactose is depleted, and the strain with the faster growth rate, **Er**, takes over (**$\mu_{Er} > \mu_{Em}$, see Fig. 2b**). Smr's amino acid production rates also affect coexistence. We measured whether Em is able to increase in frequency from rare across a range of arginine production rates, keeping the methionine production rate fixed at 1 and Smr's growth rate at 0.5. When Smr grows more slowly but produces arginine at a rate four times faster than methionine, Em is not able to invade from rare (Fig. 4b). This is because Er is no longer arginine-limited, **and it has a faster maximum growth rate than Em** (amino acid dynamics are similar to inset plots in Fig. 4a). In sum, these results suggest that coexistence via amino acid limitation can be created either by a low Smr growth rate or low amino acid production rates.

• In the paragraph beginning on line 182, it is again not clear that growth rate is the reason for these simulation results. The description makes it seem like the model has an asymmetry for no apparent reason, and a more nuanced discussion is warranted to explain why changes in amino acid depletion change the outcomes.

We have clarified that Er has a faster maximum growth rate. The relevant sentences now read, with additions in bold:

Line 211: At low arginine depletion rates, both amino acids are abundant throughout growth, and Er is able to grow more quickly and outcompete Em (**$\mu_{Er} > \mu_{Em}$, see Fig. 2b**). These

results support the conclusion that amino acid limitation for the initially-common *E. coli* strain is necessary for coexistence. When amino acids are either limiting for both strains or abundant for both strains, coexistence is lost and **Er takes over due to its faster growth rate.**

Line 218: Next, we used the model to explore coexistence in a scenario in which the *E. coli* strains deplete both amino acids. In our laboratory system, it is possible that Em depletes arginine and that Er depletes methionine at low rates. We again started Em rare and tracked its change in frequency under different depletion rates of the other *E. coli* strain's required acid. We found that depletion of arginine by Em has no effect on coexistence (Supplementary Fig. 11a), presumably because arginine only becomes more limiting for Er at early. However, if Er depletes methionine at high rates (e.g. at 90% of the rate at which Em depletes methionine), Em is unable to invade and **Er takes over due to its faster maximum growth rate** (Supplementary Fig. 11b). If both strains are able to deplete the other strain's amino acid, the two effects cancel out and coexistence is always possible, except when both strains deplete the other amino acid at the same per capita rate at which the auxotroph depletes that amino acid (Supplementary Fig. 11c). **In that scenario, Er takes over due to its faster maximum growth rate.**

• Also about this paragraph: the section about the version of the model with only one amino acid consumed by both E. coli strains would be better as a separate paragraph. And it should be explained why three species cannot coexist in a three-resource environment, given the right parameters. The text describes how changing the growth rates does not allow coexistence, but does not describe changing the Monod constants—why won't trade-offs between growth rates and Monod constants allow for coexistence of three species in three resources?

We now discuss the single amino acid scenario in a separate paragraph. We also clarify that competitive exclusion is due to unequal growth rates, and note that K_m vs. μ_{max} differences could promote coexistence (as shown in other studies), but our model involves equal K_m values. The relevant new text is shown in bold.

Line 232: We explored complete overlap in amino acid consumption by creating a model in which both *E. coli* strains require and consume the same amino acid and *S. enterica* only produces this single amino acid (Supplementary Fig. 12a). In this situation, the two *E. coli* strains compete both for lactose and for the amino acid, and the slower-growing *E. coli* strain is outcompeted by the faster growing strain, even if *S. enterica*'s growth rate is low (Supplementary Fig. 12b-d). **Consistent with previous simulations, we kept the Monod constants for lactose (e.g. $k_{Em} l_{cts}$) and amino acid equal for both *E. coli* strains. As shown elsewhere, unequal Monod constants can allow two competitors to coexist when competing for two essential nutrients²⁰. However, in a system like ours where competitors are limited by equivalent resource concentrations, coexistence is not possible with complete overlap in benefits provided by the shared mutualist.**

• *In the paragraph beginning on line 202, is there any reason to name the third amino acid in the simulation? Mentioning phenylalanine implies that it has a unique property, or even that you did another experiment with a third strain. If there is nothing special about phenylalanine, it should be renamed as a generic third amino acid to avoid confusion.*

We appreciate this suggestion, and have changed the name of the hypothetical third amino acid to “x.”

Finally, the flow of the manuscript (and particularly the Results section) is not ideal. The transition from experiments to simulations is not clear, and the reader might think that simulation descriptions are referring to experiments (the simulation figures should also be clearly labeled, either in the figure title or the image).

The experimental and modeling sections are labeled with the headers *Laboratory Experiments* and *Ecological Modeling*. To further clarify which results are derived from lab experiments vs. modeling, we’ve made sure that each paragraph of the modeling section mentions that the results are from the model. For example,

Line 156: Consistent with our laboratory system, **the model shows** that Er outcompetes Em when the two strains are grown in a lactose environment without Smr and with excess amino acids (Fig. 3a).

Line 165: To understand why Em and Er coexist despite lactose competition, we focused on nutrient dynamics **in the model**.

Line 181: To explore the importance of amino acid limitation for coexistence, **we used the model to investigate** the influence of Smr’s growth rate and amino acid production rates.

Line 204: **Another model parameter** that we hypothesized could affect amino acid limitation is the rate at which the E. coli strains deplete their amino acids (Supplementary Fig. 10).

Line 218: Next, **we used the model to explore** coexistence in a scenario in which the *E. coli* strains deplete both amino acids.

All other paragraphs already contained a sentence stating that the results pertain to the model.

We have edited the figure legends so that it is clear that Fig. 2 presents lab results, while Figs. 3-5 show modeling results.

Minor issues:

• *In some cases, adding a final sentence to the paragraphs in the Results section that sums up the findings (and/or say what they mean) could make the text more clear.*

We have added summary sentences to several paragraphs. For example:

Line 200, at the end of the Smr growth rate and amino acid production rate paragraph:

In sum, these results suggest that coexistence via amino acid limitation can be created either by a low Smr growth rate or low amino acid production rates.

Line 213, at the end of the *E. coli* amino acid depletion rate paragraph:

These results support the conclusion that amino acid limitation for the initially-common *E. coli* strain is necessary for coexistence. When amino acids are either limiting for both strains or abundant for both strains, coexistence is lost and Er takes over due to its faster growth rate.

• *Fig S4 is hard to interpret, which makes the argument for coexistence in the 3-species community less persuasive. Maybe adding initial abundances to the plot would help.*

Thank you for this suggestion. We have changed the format of the figure, added initial abundances, and included data from the new batch of this experiment. As described in the updated figure legend for this figure (which is now Supplementary Figure 6), the panels show the initial Em frequencies, and the point shapes indicate the experimental batch.

Jeff Gore

Reviewer #2:

Review of "Limitation by a shared mutualist promotes coexistence of multiple competing partners" by Hammarlund et al.

In their manuscript, the authors explore to what extent a shared mutualist partner facilitates the coexistence between two species competing for common resources. The manuscript is made of two parts: an experimental part and a modelling part. I particularly enjoyed seeing the interplay between the experiment and the model parts and, the authors explaining the experimental results based on a mechanistic model. As a theoretician, I cannot assess whether the

experiment was performed carefully, but the design was done properly for answering the questions. My review will focus on the modelling part.

Although I enjoyed reading the manuscript, I found inconsistencies in the theoretical part. It thinks it can be corrected, but this will require a substantial amount of work and literature reading.

My first point is about the “Competitive exclusion principle”. This principle has been largely overstated in ecology. This is mainly due that people tend to forget under which assumptions such a principle is valid. Actually, it holds when there is no other interactions between and within species than the “indirect” interaction of competition due to the single common resource. As soon, as there is another source of intraspecific or interspecific interaction (we can simply thin on space limitations or direct encounter among individuals) the principle doesn’t hold, and there can be enough niche differentiation to make species coexisting. The authors should consult the work of Claude Lobry, such as “Persistence in ecological models of competition for a single resource” (doi:10.1016/j.crma.2004.12.021) or his two books: chapter 3.6 of “The Consumer–Resource Relationship“ and chapter 3 of “The Chemostat”

We certainly agree that there are many mechanisms that can maintain diversity in the face of interspecific competition. We highlight that initially our system did seem to fall under the conditions that would lead to competitive exclusion, as carrying capacity for each *E. coli* isolate was set solely by lactose (both amino acids were in excess when *E. coli* growth stopped), and as the strains are isogenic other than the auxotrophy, the lactose uptake kinetics are expected to be identical. However, we attempt to account for the uncertainty around competitive exclusion by the following edits:

Line 38: When multiple species compete for the same limiting resource, one species may competitively exclude the others, leading to a loss of diversity within the community^{12,13}.

However, because species-rich communities of multiple mutualists exist in nature, certain mechanisms that maintain coexistence must exist¹⁴. (Reference 14 is Lobry et al. 2005)

Line 79: We replaced the sentence “Classically, species cannot coexist if they have different growth rates and compete for the same limiting resource” with: Growth rates are a classic measure of competitive ability in microbial systems¹⁶.

Line 283: Theoretical work predicts that multiple species typically do not coexist if they are limited by the same resource^{13,19} (though see Lobry et al. 2005). However, if the species are limited by different resources, they can more easily coexist^{22,23}.

My second point is about the invasion criterion for coexistence. Again this is a principle that has been largely overstated in ecology and its validity is not as general as it is claimed by several

authors. Actually, the invasion criterion is mathematically proven in the case of two competing species described by a Lotka-Volterra model, i.e.

$$dN_1 / dt = N_1 (r_1 - \alpha_{11} N_1 - \alpha_{12} N_2)$$

$$dN_2 / dt = N_2 (r_2 - \alpha_{21} N_1 - \alpha_{22} N_2)$$

In this model, if species 1 can invade the monoculture equilibrium of species 2 and vice versa, then both species coexist. Then this criterion has been thought, without justification, to be true whatever the number of species is and the dynamic model is. Recently in Saavedra et al. ("A structural approach for understanding multispecies coexistence", doi: 10.1002/ecm.1263) have shown that coexistence, with 3 species and a Lotka-Volterra model, can be reached without the invasion principle to be fulfilled (figures 6, 7, and 9). Thus, as stated in the manuscript I'm not a priori convinced that the invasion criterion is equivalent to coexistence in the batch model proposed by the authors on page 5 of their manuscript. The authors have to prove that it holds for their specific model, if not it is pure speculation.

We thank the reviewer for pointing out these important considerations.

- 1) Saavedra et al. demonstrate the caveat that coexistence in three member systems is possible in the absence of mutual invasibility. This caveat relies on non-transitive interactions such that no two species coexist, and indeed breaks down if species pairs do coexist. We argue the opposite of the situation covered by this caveat--that mutual invasion of competitors in a three member system is consistent with coexistence. Further we have previously shown that cross-feeding *E. coli*/*S. enterica* pairs do coexist, thereby demonstrating our system does not fall under the caveat addressed by Saavedra et al.
- 2) As described below, we now include simulated transfers to further demonstrate that our model converges on equilibrium frequencies for all three strains. This data is presented in Supplementary Figure 7.
- 3) Finally, in the chemostat model in the supplementary information we show in sections 3.2 and 3.3 that i) parameters for a 3-species steady state exist, ii) each *E. coli* strain can coexist with *S. enterica* and (iii) if the two strain *E. coli*/*S. enterica* coexistence steady states (E_m/S_{mr} and E_r/S_{mr}) are unstable to invasion by the other *E. coli* strain, then the 3-strain coexistence steady state exists. Therefore, for the chemostat model of the three strain coexistence, we recover the result for two competing species Lotka-Volterra model where the mutual invasibility does imply 3-species coexistence.

At all, I find it is a very interesting study that may deserve publication if the authors adjust and correct their theoretical part.

Thank you.

Reviewer #3:

First of all, I am sorry this review took me so long. These have been extraordinary circumstances.

This paper explores a classical question in ecology: how can two or more species coexist if they compete for the same resources? In this case, however, these two competing species share a mutualist partner. It is then, the temporal variation created by interactions within this mutualism that allows for coexistence. If two (or more) species obtain different resources from the same partner, then, they can coexist even if they compete for all other resources, as long as the shared partner depends on the mutualists to grow and does not grow too fast. This paper is a very neat example combining experiments and models and has wide implications for thinking on how different kinds of interactions affect community assembly and stability.

Thanks.

This paper proposes a mechanism of coexistence, that to my knowledge has not been applied to multispecies mutualisms. As such, I think this is a paper that inspires multiple avenues of research (e.g. How common is this mechanism in nature? How evolutionary stable is this mechanism? Under which conditions is it stable?).

*I have two larger concerns/questions, but I think they are not hard to address. First, I am not sure that the results from the mutual invasibility experiment hold-up when you have multiple species, and the community is not allowed to reach equilibrium (although from your model I think your conclusions hold-up). The problem is that you could have added initially a low density of *Salmonella* (such that it exacerbates the temporal distribution of nutrients) but over a few transfers, as the species change in relative abundance, you might increase the density of *Salmonella* making cross-feeding resources less limiting and reducing the effect of temporal variation. Have you done competition experiments with multiple transfers to evaluate what are the relative frequencies of the three strains at equilibrium? Am I missing something?*

We agree that this is an important consideration. As described above in response to reviewer 1's comments, we attempted to do multiple transfers, but we unfortunately found that our Er strain rapidly evolves arginine prototrophy. However, we believe that our data from one growth period demonstrates that Smr's equilibrium relative abundance within the community would be around 30%, and that this abundance promotes coexistence of Em and Er.

We performed an additional batch of mutual invasibility experiments in lactose media (see updated Fig. 2e), and now have data from six different initial Smr relative abundances (two batches with three different initial Em frequencies, where Smr's relative abundance varied slightly among the different Em frequencies, with a constant Smr density and slight differences in total *E. coli* strain density). When we plot Smr's initial and final frequencies, we see that Smr's frequency converges to ~0.3.

This frequency is consistent with data from our lab from a similar cross-feeding community. When the Em strain is paired with a strain of *S. enterica* that only produces methionine (the ancestor of Smr), we find that *S. enterica* reaches an equilibrium frequency of ~25% (Harcombe et al. 2014; see Fig. 3c). An intermediate frequency of *S. enterica* has also been observed in other experiments with this system (Hammarlund et al. 2019).

We believe that this indicates that Smr's equilibrium relative abundance in our three-strain community would be stable at around 30%, and would support coexistence of Em and Er.

The figure above is presented as Supplementary Figure 5, and is described briefly in the Results section at line 121.

Smr also increased in frequency from rare and decreased when initially common (Supplementary Fig. 5), consistent with previous results showing that *S. enterica* stably coexists with cross-feeding *E. coli*.¹⁹

Additionally, to further explore the importance of Smr's abundance, we performed transfer experiments in the model across a range of initial Smr densities. During each growth cycle, the community grew until equilibrium was reached, and then was diluted 100-fold. We found that Smr stabilizes at 50% of the community and that Em and Er coexist regardless of the initial Smr frequency. In the figure below, the three columns show three initial densities of Smr, with the default starting density of 100 shown in the middle. The rows show two different initial Em frequencies (0.1 and 0.9). Regardless of its initial frequency, Smr reaches and maintains a frequency of 0.5 (Smr / total community). Em stabilizes at 50% of the *E. coli* portion of the community (Em / (Em+Er)) both when started at 10% and at 90%.

This figure is shown as Supplementary Figure 7, and discussed briefly in the Results section in a new sentence at line 161:

Stable equilibrium frequencies are quickly reached when communities are transferred with 100-fold dilutions, and coexistence is robust to changes in initial Smr densities (Supplementary Fig. 7).

Second (and I feel some hesitation about bringing this up given COVID-19) I felt like the experimental observation was a great motivation for the model. However, the model makes some interesting predictions and I was left hoping for some of those to be evaluated experimentally. For example, it might not be possible to change the growth rate of Salmonella, but it would be interesting to know how coexistence changes across a gradient of Salmonella initial density (prior to equilibration), or measure the growth rates of the E. coli strains when grown in the supernatant of Salmonella after different periods of growth in the presence of each

of the E.coli strains. These experiments would, in my opinion, make the paper stronger, but they are not necessary.

We also wish we were able to experimentally alter the growth rate and the amino acid production rates of Smr, but both practical and COVID-related constraints make this difficult. We hope that the lab data and modeling data shown in response to the previous comment help to demonstrate that coexistence is robust to different initial Smr frequencies.

Finally, I had a couple of smaller comments and questions:

P2-L32: Most bees get both pollen and nectar from flowers. I would check this to avoid making pollinator biologists mad.

We agree that the bee pollen/nectar example was imperfect. We've changed the example to a shared mutualist arbuscular mycorrhizal fungal species that provides different resources and/or services to two different plant hosts. AM fungi provide services like protection against biotic and abiotic stressors, and resources like phosphorus, nitrogen, and micronutrients. In a hypothetical scenario that we now describe in the introduction and in the legend of Figure 1, a shared mutualist AM fungal species provides pathogen protection to one partner mutualist plant, and phosphorus to a second partner mutualist plant. Both plants excrete carbon compounds that the fungi consume, and the plants compete for water, analogous to lactose competition in our system.

P5- Er and Em equations: I am not sure if I completely understood why were the consumption of lactose and methionine or arginine terms multiplied and not added. Is there co-utilization?

Adding the two terms would suggest that they were replaceable. By multiplying, if either goes to 0, there is no growth, which is true for our system in the lab where amino acids and lactose are both essential. One could argue for using the minimum of either rather than the product, but initial simulations show that using the minimum doesn't affect our results.

P5-L145: I also was not sure why did you use these growth rates instead of the ones you measured?

We performed several of the modeling experiments before we had carefully measured growth rates in the lab, so we approximated growth rates based on preliminary data. Because additional features of the model were not parameterized based on lab data, we view the model as only qualitatively representative of our system, and don't expect that using growth rates measured in the lab would alter our findings.

P6-L183: “We found that the rate at which *Em* depletes methionine has no effect on coexistence” - That is interesting! Why?

When *Em* begins in the minority, methionine is plentiful and *Em* grows at its maximum growth rate, while *Er* is limited by low arginine concentrations. Across the range of methionine depletion rates that we originally had tested, higher methionine depletion by *Em* (relative to arginine depletion by *Er*) had no effect on coexistence, because methionine was still more abundant than arginine at early timepoints despite high depletion rates.

However, we’ve now tested a larger range of *Em* methionine depletion rates, and found that coexistence is lost above a value of 9 (where methionine depletion by *Em* is 9x greater than arginine depletion by *Er*). Here, *Em* depletes methionine so fast that its growth rate becomes limited by methionine. Both strains are limited by their amino acids, and *Er* outcompetes *Em* due to its faster growth rate.

We have updated part a of the figure (now SI Fig. 10) to show this greater range for the *c Em met* parameter, and updated the figure legend and the main text to discuss this result.

The updated paragraph in the results section at line 204 reads:

Another model parameter that we hypothesized could affect amino acid limitation is the rate at which the *E. coli* strains deplete their amino acids (Supplementary Fig. 10). To test for coexistence, we started Em rare (10%) and tracked its change in frequency under different amino acid depletion rates. We found that Er competitively excludes Em when Em consumes methionine rapidly, at nine times the rate at which Er consumes arginine (Supplementary Fig. 10a). Coexistence is lost because methionine becomes limiting for Em, and Er is able to grow faster. Coexistence is also lost when Er's arginine depletion rate is low (around 25% of the default rate; Supplementary Fig. 10b-c). At low arginine depletion rates, both amino acids are abundant throughout growth, and Er is able to grow more quickly and outcompete Em ($\mu_{Er} > \mu_{Em}$, see Fig. 2b). These results support the conclusion that amino acid limitation for the initially-common *E. coli* strain is necessary for coexistence. When amino acids are either limiting for both strains or abundant for both strains, coexistence is lost and Er takes over due to its faster growth rate.

P8-251: What if strains compete for the same resource in the mutualism but different external resource? I think this parallels many mutualisms in nature, and it would be interesting to know (maybe not in this paper?) under which conditions is this possible.

That is an interesting question! A similar scenario is presented in Fig. 1C-D in Jones et al. 2012 (*The fundamental role of competition in the ecology and evolution of mutualisms*). There, the authors speculate that the partner mutualists' quality (the degree of benefits given to the shared mutualist) and competitive ability for both external resource(s) and the shared mutualist-provided resource may determine whether coexistence is possible. Partner choice, which is not possible in our system, may also determine coexistence or competitive exclusion.

Other authors have taken modeling approaches to explore competition for a shared mutualist-provided resource. However, to our knowledge, those studies don't explicitly explore consumption of separate external resources.

In a system similar to ours with no partner choice and equal partner mutualist quality, we predict that coexistence will be determined by which resources the competing strains are limited by. If the better competitor for the mutualist-provided resource is limited by its external resource rather than the mutualist-provided resource, there should be coexistence. However, if the mutualist-provided resource is limiting for both competitors, the stronger competitor should take over. It would be interesting to test this prediction with our model in the future, and it would be feasible to set up such a lab system as well.

Fig 3a (and S9) Why is there a little dip in the change of Em frequency at intermediate frequencies of Em?

The dip is the result of plotting absolute change in frequency. When E_m starts at 1%, the most it is able to decrease is 1%. When it begins at 50%, there is a greater frequency increase of the better competitor (E_r) because there's more room for change. When E_m begins at 99%, even if E_r doubles in frequency, there will still only be a 1% reduction in the frequency of E_m . With our default parameters, the species only grow ~10-fold in one growth cycle, so frequency changes are slight at all starting frequencies, but they are greatest when starting at 50%.

We thank all of the reviewers for the thoughtful critiques and suggestions. Please contact us if we can elaborate on our responses or provide more information.

- Sarah Hammarlund, Ross Carlson, Tomáš Gedeon, and Will Harcombe

Reviewers' Comments:

Reviewer #2:

Remarks to the Author:

I'm satisfied with the revision. I recomande the manuscript for publication.

Reviewer #3:

Remarks to the Author:

Thank you for your thoughtful responses. My concerns have all been addressed. I think the manuscript is clearer and I enjoyed reading it again. It was great to see that starting from different frequencies of Smr you get convergence in the final frequency. I think this strongly suggests that communities are close to equilibrium within one transfer or so. My only comment is to add (as supplementary material) the data from the multi-transfer competition experiments (I do not think this is completely necessary, but I think it really strengthens your argument, and it seems like you already have the data). You should mark in this figure that after 3-4 days you observed reversals to prototrophy, altering the outcome of the competition. Given that you seem to get stability fairly quickly in your model, this should show that stability is also achieved rapidly in the experimental model.

Very nice work,

María Rebolleda-Gómez

Reviewer #3:

Thank you for your thoughtful responses. My concerns have all been addressed. I think the manuscript is clearer and I enjoyed reading it again. It was great to see that starting from different frequencies of Smr you get convergence in the final frequency. I think this strongly suggests that communities are close to equilibrium within one transfer or so. My only comment is to add (as supplementary material) the data from the multi-transfer competition experiments (I do not think this is completely necessary, but I think it really strengthens your argument, and it seems like you already have the data). You should mark in this figure that after 3-4 days you observed reversals to prototrophy, altering the outcome of the competition. Given that you seem to get stability fairly quickly in your model, this should show that stability is also achieved rapidly in the experimental model.

Thank you for the suggestion. We have added this figure:

With legend:

Figure 5 | Competitive exclusion and coexistence over multiple transfers. a , In lactose + methionine + arginine media, Er outcompetes Em over multiple growth cycles (Batch C, square points). **b** , In lactose media with Smr, Em is maintained at an intermediate frequency (Batch D, square points). Cultures were diluted 100-fold into fresh media after each two-day growth period, and plated to count colonies. At the timepoint 5 plating, we observed a high frequency of prototrophic colonies derived from the Er strain, suggesting that prototrophic mutants arose and increased to a high frequency during the experiment. To be conservative, we recommend disregarding data after timepoint 2. The evolutionary lability of this system makes mutual invasibility experiments over one growth cycle a better method to assess coexistence.

We refer to this figure in the main text at lines 107 and 130.

We thank all of the reviewers for their thoughtful critiques and suggestions throughout the review process.

- Sarah Hammarlund, Ross Carlson, Tomáš Gedeon, and Will Harcombe